# Alteration of Serum Proteome in Levo-Thyroxine-Euthyroid Thyroidectomized Patients

**DOI:** 10.3390/jcm11061676

**Published:** 2022-03-17

**Authors:** Claudia Landi, Silvia Cantara, Enxhi Shaba, Lorenza Vantaggiato, Carlotta Marzocchi, Fabio Maino, Alessio Bombardieri, Alfonso Carleo, Fabrizio Di Giuseppe, Stefania Angelucci, Luca Bini, Maria Grazia Castagna

**Affiliations:** 1Functional Proteomics Laboratory, Department of Life Sciences, University of Siena, 53100 Siena, Italy; enxhi.shaba@unisi.it (E.S.); lorenz.vantaggiato@student.unisi.it (L.V.); luca.bini@unisi.it (L.B.); 2Department of Medical Surgical and Neurological Sciences, University of Siena, 53100 Siena, Italy; cantara@unisi.it (S.C.); carlottamarzocchi@libero.it (C.M.); fabio.maino@ao-siena.toscana.it (F.M.); a.bombardieri@student.unisi.it (A.B.); mariagrazia.castagna@unisi.it (M.G.C.); 3Department of Pulmonology, Hannover Medical School, 30625 Hannover, Germany; alfonsocarleo@yahoo.it; 4Proteomics Unit, Department of Medical, Oral and Biotechnological Sciences, University “G. d’Annunzio” Chieti-Pescara, 66100 Chieti, Italy; f.digiuseppe@unich.it (F.D.G.); s.angelucci@unich.it (S.A.)

**Keywords:** proteomics, acquired hypothyroidism, reduced FT3, LT4, TSH

## Abstract

The monotherapy with levo-thyroxine (LT4) is the treatment of choice for patients with hypothyroidism after thyroidectomy. However, many athyreotic LT4-treated patients with thyroid hormones in the physiological range experience hypothyroid-like symptoms, showing post-operative, statistically significant lower FT3 levels with respect to that before total thyroidectomy. Since we hypothesized that the lower plasmatic FT3 levels observed in this subgroup could be associated with tissue hypothyroidism, here we compared, by a preliminary proteomic analysis, eight sera of patients with reduced post-surgical FT3 to eight sera from patients with FT3 levels similar to pre-surgery levels, and six healthy controls. Proteomic analysis highlights a different serum protein profile among the considered conditions. By enrichment analysis, differential proteins are involved in coagulation processes (PLMN-1.61, -1.98 in reduced vs. stable FT3, *p* < 0.02; A1AT fragmentation), complement system activation (CFAH + 1.83, CFAB + 1.5, C1Qb + 1.6, C1S + 7.79 in reduced vs. stable FT3, *p* < 0.01) and in lipoprotein particles remodeling (APOAI fragmentation; APOAIV + 2.13, *p* < 0.003), potentially leading to a pro-inflammatory response. This study suggests that LT4 replacement therapy might restore biochemical euthyroid conditions in thyroidectomized patients, but in some cases without re-establishing body tissue euthyroidism. Since our results, this condition is reflected by the serum protein profile.

## 1. Introduction

Patients undergoing thyroidectomy need to maintain correct thyroid hormone (TH) levels by replacement therapy with levo-thyroxine (LT4) [1]. The rationale for the use of LT4 is supported by the fact that the peripheral desiodase-mediated conversion of T4 into T3 is able to guarantee a correct homeostasis of THs in target tissues [2]. In the majority of patients, LT4 therapy is effective. However, in a subgroup of thyroidectomized subjects [3,4], typical symptoms of hypothyroidism persist with a reduced quality of life (QoL) despite normalization of serum TSH levels.

We recently reported that a subgroup of athyreotic patients treated with LT4 therapy showed post-operative, statistically significantly lower FT3 levels (reduced FT3) with respect to those before total thyroidectomy [5]. We also demonstrated that the lower plasmatic T3 level found in 34% of the analyzed patients was associated with the X/Ala D2 genotype, which suggests that these patients do not respond optimally to standard LT4 therapy.

Little is known about the relationship between TH levels and circulating proteome profile. In the literature, only a few studies evaluated the changes in circulating proteome in patients undergoing LT4 replacement therapy [6]. Among these, two papers reported plasma proteomic analyses in an experimentally induced human thyrotoxicosis model [7,8]. Engelmann et al. explored the effects of T4 excess and found a positive correlation between T4 levels and 10 proteins involved with blood coagulation [7]. Pietzner et al. detected 63 proteins significantly associated with serum FT4 levels. Among these, complement and coagulation proteins were increased and apolipoproteins were decreased [8]. Only one study explored plasma proteome changes that occur in the transition from hypothyroid to euthyroid status in patients during LT4-therapy [9]. This comparison revealed alterations of proteins involved in acute phase response, highlighting that LT4 reduces pro-inflammatory cytokine levels. No studies took into consideration the effects of LT4 on circulating proteome in thyroidectomized patients.

According to our previous results [5], we hypothesized that the lower plasmatic FT3 levels observed in a subgroup of athyreotic patients treated with LT4 therapy, could be associated with tissue hypothyroidism. To explore this hypothesis, we performed a preliminary proteomic analysis on a selected population of eight sera from patients with reduced post-surgery FT3 and eight sera from patients with stable post-surgery FT3 levels. Moreover, sera from six euthyroid subjects and without comorbidities were considered as a healthy control group for the comparison. Dysregulated protein spots, found by proteomic, were considered for principal component and heatmap analyses. Identified proteins were submitted to enrichment analysis in order to highlight their position into specific molecular pathways. Interesting proteins and their behavior were validated by two-dimensional western blotting in an alternative cohort of samples to confirm their particular pattern and their amount in reduced FT3.

## 2. Materials and Methods

### 2.1. Patients’ Population

Patients were selected from the study population evaluated in a previously published study [5] and included patients submitted to total thyroidectomy. Inclusion criteria were: (1) thyroid profile data obtained within 10 months of surgery; (2) post-surgical thyroid profile obtained at least 6 months after achievement of a stable thyroid hormone status on LT4 therapy; (3) pre- and post-surgical serum TSH levels not different by more than ±0.5 mIU/L. Patients with an abnormal thyroid profile (hypo- or hyperthyroidism) before surgery, patients with comorbidities or receiving drugs that can interfere with thyroid function, and patients affected by malabsorption-related conditions were excluded.

We arbitrarily selected a change of at least 0.5 pg/mL as a significant variation between pre- and post-surgical FT3 value. According to this criterion, patients were classified as having “reduced FT3” when post-surgical FT3 levels were at least 0.5 pg/mL lower than pre-surgical FT3 values. In all patients, FT3 levels were into the normal range before and after surgery and no difference between pre- and post-surgical TSH levels were observed [5]. From our previous cohort, we selected postsurgical sera of 16 patients (8 patients with “reduced FT3” and 8 with “unchanged FT3”). The study cohort included 50% of females with a mean age of 50.25 ± 14.9 years (range 24–80 years) submitted to total thyroidectomy for nodular disease. At final histology, 14/16 (87.5%) patients had differentiated thyroid carcinoma and 2/16 (12.5%) had a benign goiter. Immediately after surgery, patients were treated with LT4 to obtain comparable pre-surgical TSH levels, with a mean dose of 114.5 µg/day and a mean dose/kg of 1.6 µg of LT4. Fasting blood samples were collected at 08.00–09.00 h before patients assumed the LT4 tablet, and all determinations were performed with a chemiluminescent immunometric assay (Access Immunoassay Systems 2006, Beckman Coulter, Milan, Italy). Normal ranges were 2.5–4.5 pg/mL for FT3, 5.8–16.4 pg/mL for FT4, and 0.4–4.0 mU/L for TSH. Control subjects were matched for age and gender. They had no history of concomitant pathologies and were not on any therapy.

All methods and procedures were carried out in accordance with relevant guidelines and regulations. The study was approved by our local ethical committee “Office of Ethical Affairs. AziendaOspedaliero-UniversitariaSenese” (protocol #18923). Informed consent was obtained from all subjects enrolled in the study.

### 2.2. Proteomic and Enrichment Analysis

Blood was collected directly into serum tubes (BD vacutainer, SST II Advance, Plymouth, UK) and centrifuged for 10 min at 1690× *g*. The serum was recovered and stored at −80 °C until proteomic analysis. Samples were singularly prepared and resolved by two-dimensional electrophoresis (2DE), according to Landi C et al. [10]. MALDI-ToF mass spectrometry by Peptide Mass Fingerprint (PMF) was considered for protein identification [11]. The mass spectrometry proteomics data have been deposited to the ProteomeXchange Consortium via the PRIDE [12] partner repository with the dataset identifier PXD028378. The data analysis of relative percentage of spot volume was carried out using the Differential Analysis tool in XLStat software (XLSTAT-life Science—Paris, France; date access 10 May 2021). Then, %Vol differences among matched spots, from the three investigated groups (i.e., control, reduced FT3, and stable FT3), were evaluated by a statistical non-parametric Kruskal–Wallis test (*p* ≤ 0.05). The mean rank of significant results was then compared by Dunn’s multiple comparison test. Statistically significant differences were then processed according to the ratio value ≥1.5 of corresponding %Vol means. Spots presenting a *p*-value lower than 0.05 were considered as significant and were submitted to Heatmap and Principal Component Analysis (PCA). In particular, spots and samples were clustered in the Heatmap according to Euclidean distance. The UniProt accession numbers of the identified proteins were used to perform enrichment analysis by MetaCore software (https://portal.genego.com, accessed on 7 June 2021) (Clarivate analytics, Boston, MA, USA).

### 2.3. Bidimensional Western Blot Validation of A1AT and APOAI

After bidimensional electrophoresis separation, proteins were transferred onto nitrocellulose membrane (Hybond ECL, GE Healthcare UK Limited., Amersham Place, Little Chalfont, Buckinghamshire, UK) [13,14,15]. Ponceau Red staining (0.2% *w/v* Ponceau S in 3% *v/v* trichloroacetic acid) of the membranes was performed to verify correct protein transfer and to assess equal protein loading (data not showed). Western blot analysis was performed using: goat polyclonal antibodies against alpha 1 antitrypsin from Santa Cruz Biotechnology (San Jose, CA, USA) and mouse monoclonal anti-apolipoprotein AI (Santa Cruz Biotechnology). Goat peroxidase-conjugated polyclonal anti-rabbit and anti-mouse IgG secondary antibodies were purchased from Bio-Rad, (© 2022 Bio-Rad Laboratories, Inc., Hercules, CA, USA, Bio-Rad Laboratories S.r.l., Segrate, Italy). All antibodies were diluted according to the manufacturer’s instructions and chemiluminescent signals, obtained after using an ECL kit (GE Healthcare), were captured at different time-points of exposure by exposing membranes to Hyperfilm ECL X-ray films (GE Healthcare). The digital 2D western blot images were visualized by Image Master platinum 7.0.

## 3. Results

### 3.1. Comparison of Post-Surgical Thyroid Hormone Levels between “Reduced” and “Stable” FT3 Patients

Based on our previous report [5], we selected from the group of patients with “reduced” or “stable” FT3 at post-surgical evaluation, eight patients/group to submit to proteomic analysis. Moreover, we added six sera from healthy subjects as controls. The two groups of athyreotic patients were similar for all parameters evaluated excepted for serum FT3 value at post-surgical evaluation that were significantly lower in the subgroup of “reduced FT3” patients and for a Pro-Kg LT4 dose that was significantly higher in the subgroup of reduced FT3 (Table 1).

As shown in Table 2, no significant difference between pre- and post-surgery serum TSH and FT4 levels (by Wilcoxon test for paired data) were observed in the two subgroups of athyreotic patients. The median of post-surgical serum FT3 level was significantly lower in the “reduced” group (*p* = 0.03; 3.0 pg/mL versus 3.75 pg/mL) when compared with pre-surgical values, although in the reference range. No differences in FT3 levels between pre- and post-surgical evaluation were observed in the subgroup of “FT3 stable” according to the inclusion criteria.

### 3.2. Proteomic Analysis Results

The analyzed dataset enumerated ~1200 protein spots per gel. The unsupervised analysis (data not shown), performed as data quality control, was obtained by analyzing the top 100 spots with higher variance intensity through unsupervised heatmap and PCA and the results did not highlight any outliers nor clustering related with any possible covariant disturbances.

However, the comparison emphasized 57 significant differentially abundant spots (DAS) between reduced FT3 and stable FT3 groups (Table 3). The 57 differential spots are showed in the reference gel maps in Figure 1.

The PCA carried on the 57 DASs summarized the 86.79% (PC1:69.85% and PC2:16.93%) of the variance and the three groups clustered alongside the second component (Figure 2). In addition, Figure 2 showed the contributions of each significant variant in the first two PCs, highlighting spots 45 and 50 as the most significant.

Once protein identification by MALDI-ToF mass spectrometer was performed, 34 proteins were identified. Mascot Search Results were reported in Table 3. Interestingly, some identifications highlighted characteristic protein isoform distributions on 2DE gels. Highly abundant spots 34, 35, 45, 50 in reduced FT3 were all identified as full-length apolipoprotein AI. In particular, spots 45 and 50, considered as the most significant by PCA, showed a mixture with a fragment of A1AT, suggesting a colocalization of these two protein species on 2DE gel (Table 3, Figure 1 and Appendix A). Instead, among low abundant spots in reduced FT3, there were spots 43, 44, and 39, identified, respectively, as a mix of APOAI and A1AT, N-terminal, and C-terminal fragments of APOAI (Table 3, Figure 1 and Appendix A). In order to understand if the APOAI and A1AT mix was a physiopathological condition or a contamination, we performed a two-dimensional western blot (2DWB) analysis. Figure 3 reports 2DWB images of APOAI protein pattern whose results are in line with proteomic results. Spots 34, 35, 45, 50, identified as full-length APOAI, were confirmed to be more highly abundant in reduced FT3. Only spot 43 was shown as APOAI in 2DWB, confirming its lower abundance in FT3 reduced condition.

Instead, Figure 4 reported A1AT 2DWB analysis, and spots 43, 45, and 50, identified as a mixture of a A1AT fragment and APOAI (Table 3), confirm the presence of A1AT together with APOAI. Two-dimensional western blotting also showed that A1AT in spots 43, 45, and 50 is lowly abundant in reduced FT3, revealing that the signal observed in 2DE is in major part produced by overlapping APOAI.

Similarly to PCA, the heatmap branching showed in Figure 5 separated the samples into two principal clusters corresponding to controls and thyroidectomized patients. The thyroidectomized group of patients was further separated into two other distinctive groups corresponding to stable and reduced FT3, where stable FT3 samples clustered near CTRL samples.

### 3.3. Enrichment Analysis Results

Enrichment analysis by MetaCore software built the network of molecular interactions related to the identified proteins. Figure 6A shows the protein network where plasmin, alpha1-antitrypsin, APOE, plasminogen, and APOA1 were central functional hubs, i.e., proteins with a higher number of interactions with respect to the other proteins. Light blue lines in bold in the interactome show the canonical paths delineated in Appendix A that also reports their relative GO biological processes.

Results by process network analysis in Figure 6B showed that the inflammation, induced by complement system, IL-6 signaling, kallikrein-kinin system, and innate inflammatory response, represents the central processes where the identified proteins are involved (Appendix A).

Additionally, Pathway Maps analysis (Figure 6B and Appendix A) highlights a strong involvement of the complement, lipid particle remodeling, and blood coagulation with the terms “immune response by classical, alternative, lectin-induced complement pathways”, “alternative complement cascade disruption in age-related macular degeneration” and “complement pathway disruption in thrombotic microangiopathy”, “blood coagulation”, and terms such as “lipoprotein metabolism”, “HDL-mediated reverse cholesterol transport”, and “HDL dyslipidemia in type 2 diabetes and metabolic syndrome X”.

Thanks to the Process Network and Pathway Maps analyses, it was possible to rebuild in Figure 6A the three principal protein networks of blood coagulation (yellow), complement system (orange), and lipoprotein particle remodeling (blue) that suggest the activation of different pro-inflammatory pathways.

## 4. Discussion

Patients undergoing thyroidectomy are normally treated with LT4 replacement therapy, but a substantial proportion of them experience hypothyroid-like symptoms despite normal TSH levels. During LT4 replacement, levels of the active hormone FT3 were reported in all patients to be in the target range, but part of them shows a slight inflection of FT3. Our previous study reported that thyroidectomized patients carrying Thr92Ala polymorphism are at increased risk of reduced serum FT3 concentrations [5,16]. We hypothesized that the lower plasmatic FT3 levels observed in a subgroup of athyreotic patients treated with LT4 therapy, could be associated with tissue hypothyroidism.

To this purpose, we performed a differential proteomic analysis of serum samples in “reduced” FT3, “stable” FT3 patients, and healthy subjects. Proteomics is used for research purposes to highlight potential biomarkers that could be used in clinical practice (translational medicine). The analysis highlighted 57 differential abundant spots. PCA separated the samples in three distinct groups, and spots 45 and 50, identified as APOAI and fragments of A1AT, have a higher relevance on these results. Heatmap analysis on proteomic results clustered our samples into two principal clusters corresponding to LT4-euthyroid thyroidectomized patients and healthy controls, highlighting that there is a substantial difference between healthy and thyroidectomized subjects. The latter group clustered into two other sub-groups, revealing a difference between serum proteomes of stable and reduced FT3, where stable is positioned next to CTRLs.

In particular, CFAH, CFAB, coagulation factor XIII B chain (F13B), vitamin D binding protein (VTDB), APOA4, C1QB, APOAI, hemoglobin, C1S, and A1AT fragments were up-regulated in reduced FT3 with respect to stable FT3 patients, while PLMN, CFAB fragments, histidine-rich glycoprotein, and some APOAI fragments were down-regulated in reduced FT3 with respect to stable FT3 patients.

On the other hand, serum down-regulated proteins in thyroidectomized patients and healthy controls were CFAH, alpha 1B glycoprotein, VTDB, APOAIV, C3 fragment C-term, C1S, and N-term APOAI fragments, while up-regulated proteins were PLMN, full-length CFAB and some fragments, serotransferrin, F13B, albumin fragments, APOAI fragments C-term, apolipoprotein E, and heavy constant gamma 2 immunoglobulin.

Enrichment analysis reported that the differential proteins identified participate in processes such as blood coagulation, complement activation, and lipoprotein particles remodeling. All these processes, if dysregulated, have the potentiality to induce a pro-inflammatory response by IL6 signaling, kallikrein-kinin system activation, and innate inflammatory response that could be considered responsible for hypothyroidism symptoms. In particular, blood coagulation mechanisms were suggested by plasminogen, coagulation factor XIII B chain, and A1AT, which were up-regulated in thyroidectomized samples than in controls, whereas plasminogen and A1AT resulted in down-regulation in a reduced respect to stable FT3. Complement activation pathways were proposed by altered levels of C3, C1qb, C1s, CFAB, and CFAH. An up-regulation of these proteins in thyroidectomized patients with respect to controls, which was maintained in reduced respect to stable FT3 patients, was observed. This suggests that different complement pathways were activated following thyroidectomy and were maintained in patients that experienced reduced FT3. In accordance with our results, Bitencourt et al. indicated that there is a connection between the immune system and thyroid hormones, since T3 and T4 hormones affect the lytic potency of the complement system [17]. In particular, we found an up-regulation of fragments that could be generated during the classical, alternative, or lectin-induced activation of the complement. These mechanisms could have important biological functions, including facilitation of phagocytosis, clearance of immune-complexes, inflammation, immune response, and tissue homeostasis [18,19]. To our purpose, process network analysis suggests that dysregulated proteins involved in blood coagulation and complement systems, such as CFAH, C3, C1s, Plasminogen, Alpha 1 antitrypsin, may be responsible of inflammatory processes by complement activation, kallikrein-kinin system (KKS) activation, and IL-6 signaling. Complement and KKS are reported to be activated during vascular inflammation [20] and their extensive activation on the endothelium promotes thrombosis, leukocyte recruitment, vascular permeability, and vascular wall injury [17,20]. Basically, the complement, coagulation, and fibrinolytic systems lead to thrombo-inflammation which collectively leads to the activation of blood cells such as polymorphonuclear cells, monocytes, platelets, and endothelial cells lining the vessels [21]. Therefore, the altered coagulation processes and complement system activation that we found could suggest the presence of a potential vascular inflammation in “reduced” FT3 patients, as already reported in subclinical hypothyroidism [22]. It is of interest that different studies associate higher serum levels of inflammatory molecules to the onset of fatigue and migraines, a characteristic symptom of hypothyroidism [23,24,25], as well as to depression [26]. Indeed, inflammation related to a potential endothelial dysfunction, as suggested by our data, was already reported in depression [27] and may also prove to be the cause for poor QoL in FT3-reduced patients.

The role of plasminogen, plasmin, and A1AT as central hubs of our protein network makes them potential biomarkers of hypothyroidism symptoms despite values of TSH in the reference range. Moreover, the characteristic protein fragmentation of A1AT suggests a protein impairment in reduced FT3 serum that could be responsible and peculiar for this condition. The low abundance of A1AT fragments in reduced FT3 not only points to an alteration in coagulation processes but also to a serine protease/serpins impairment, a potential cause of hypothyroidism symptoms. Chadarevian R et al. reported plasma levels of some components of the fibrinolytic system correlated to plasma levels of T4 [28]. This suggests a possible correlation between FT4 replacement therapy modulation and coagulation cascade.

Another interesting finding of our study regards the lipoprotein particle remodeling pathway suggested by the dysregulation of APOA4, APOAI, APOE, albumin, and haptoglobin. Because thyroid hormone influences all aspects of lipid metabolism such as synthesis, mobilization, and degradation, dyslipidemia is very common in thyroid disease [29]. In this case, APOAI and APOE, up-regulated in reduced FT3 patients, in agreement with data reported by O’Brien et al. [30], resulted in central functional hubs of our analysis suggesting their potential role as biomarkers. In particular, our experiments highlight a high abundance of the full-length APOAI. Simultaneously, we observed a decrease of APOAI fragments in reduced FT3 with respect to stable FT3, accordingly with Jung et al. [29]. This demonstrates a specific resistance to proteolytic mechanisms. Dysregulation of APOAI management could suggest a potential lipoprotein particle dysfunction. To this purpose, Jung et al. report that increased HDL-cholesterol levels with impaired function, persisted despite restoration of thyroid hormone levels. Moreover, the concentration of APOE showed a similar behavior to that of HDL-cholesterol [29], in agreement with our results. Changes in thyroid function are associated not only with changes in the concentrations of various plasma lipid components but also with changes in HDL functions [29], and our results sustained this thesis.

In light of our findings, enrichment analysis confirms that the above-mentioned pathways induce pro-inflammatory responses by complement system activation but also by the kallikrein-kinin system and by IL-6 signaling, according to Zhou et al. [31]. The latter reported an increasing of IL-6 and TNF-α after hypothyroidism induction in rats. Together, these results sustain an interrelationship between inflammation and hormone derangement [32], probably establishing a vicious circle reflected in the impaired QoL of patients at physical and neurocognitive levels.

In light of our results, since there is no evidence of the cause of hypothyroidism symptoms in FT3-reduced patients from a clinical point of view, blood coagulation levels, complement activation, lipoprotein particle composition, and levels of inflammation could be evaluated in levo-thyroxine-euthyroid thyroidectomized patients. This would permit an assessment of their modulation in a wide cohort of samples during the follow-up of treatment for a more complete clinical picture.

Limitations: further studies including a large cohort of patients will be needed to confirm our results. Moreover, a serum proteomic comparison between pre- and post-thyroidectomy in the same patients could be necessary. In order to validate our results, the level of inflammatory molecules to peripheral level, the coagulation and fibrinolytic state of the patients, as well as the lipoprotein particle functionality, should be analyzed.

## 5. Conclusions

During LT4 replacement therapy, some thyroidectomized patients show slightly reduced FT3 levels. Comparing serum proteomes, we found that in reduced FT3 patients, differential proteins strongly suggest a behavior similar to overt hypothyroidism with a differential abundance of proteins involved in complement activation, blood coagulation, and lipoprotein particle remodeling. In particular, a characteristic A1AT and APOAI proteoform profile was reported, symptomatic of a serine protease/serpins impairment as well as of an alteration in coagulation processes and changes in HDL functions. Together, these molecular ways sustain a proinflammatory response by the kallikrein-kinin system, by IL-6 signaling, and by innate inflammatory response that led to the hypothyroidism symptoms.

In conclusion, this study suggests that LT4 replacement therapy restoring euthyroid conditions might not be sufficient in all thyroidectomized patients. The differential serum protein profile found suggests molecular pathways to be checked and supports the hypothesis that the lower plasmatic FT3 levels observed in this subgroup could be associated with tissue hypothyroidism phenotype.

## Figures and Tables

**Figure 1 jcm-11-01676-f001:**
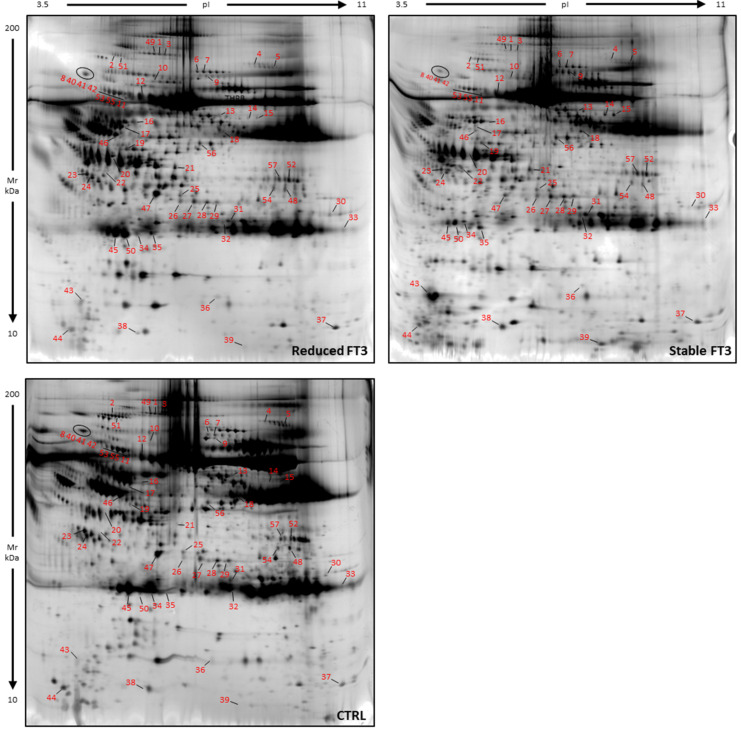
Electrophoretic maps of reduced FT3, stable FT3 samples, and control (CTRL), highlighting the differential spots found. Numbers in red correspond to that in Table 3.

**Figure 2 jcm-11-01676-f002:**
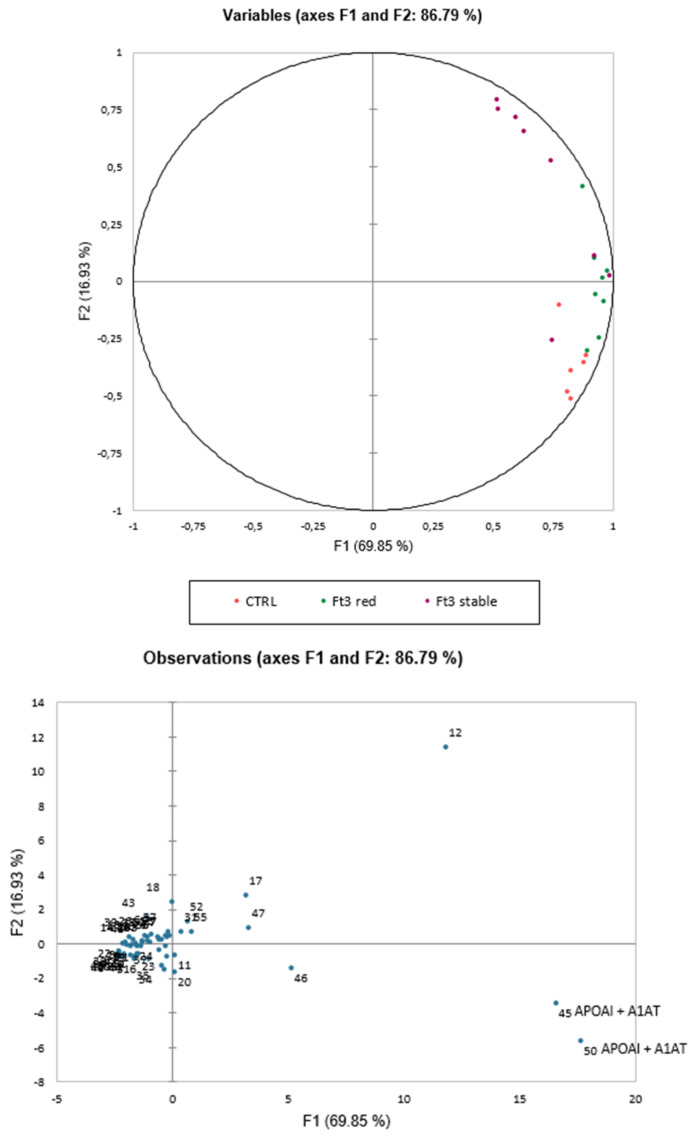
Principal component analysis (PCA) showed the 57 differential abundant spots (DASs) and summarized the 86.79% (PC1:69.85% and PC2:16.93%) of the variance and the three groups clustered alongside the second component. The image also reports the histogram with the contributions of each significant variant in the first two principal components (PCs), highlighting spots 45 and 50, identified as APOAI and fragments of A1AT, as the most significant.

**Figure 3 jcm-11-01676-f003:**
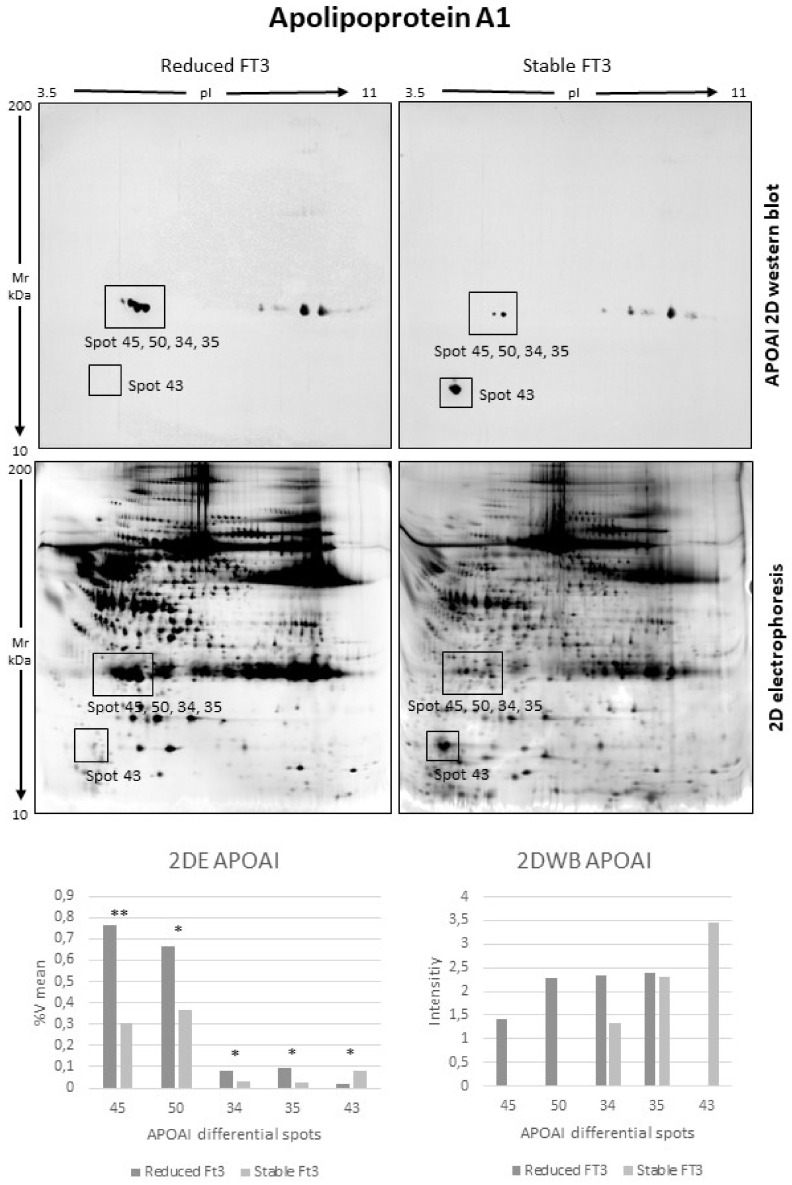
Two-dimensional western blot (2DWB) analysis of apolipoprotein AI (APOAI) (* = *p*-value < 0.05; ** = *p*-value < 0.01) (2DWBs reported in figure are the entire WB membrane images). Squares on the images highlight the areas where spots 45, 50, 34, 35, and 43 are resolved.

**Figure 4 jcm-11-01676-f004:**
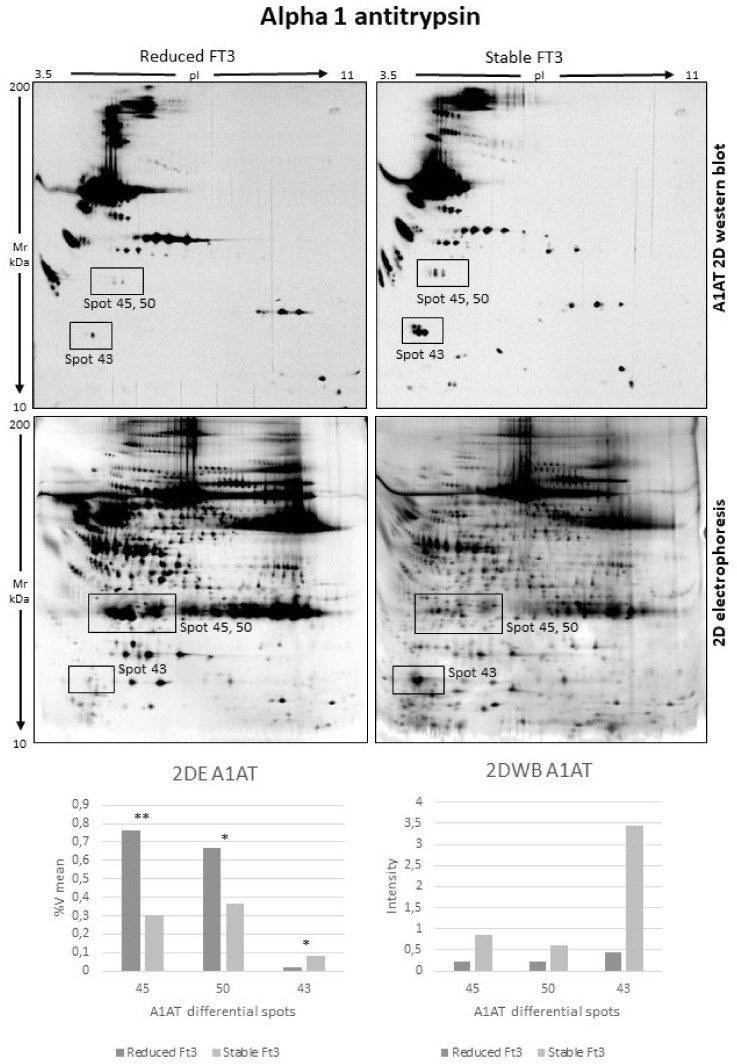
Two-dimensional western blot (2DWB) analysis of alpha 1 antitrypsin (A1AT) (* = *p*-value < 0.05; ** = *p*-value < 0.01) (2DWBs reported in figure are the entire WB membrane images). Squares on the images highlight the areas where spots 45, 50, and 43 are resolved.

**Figure 5 jcm-11-01676-f005:**
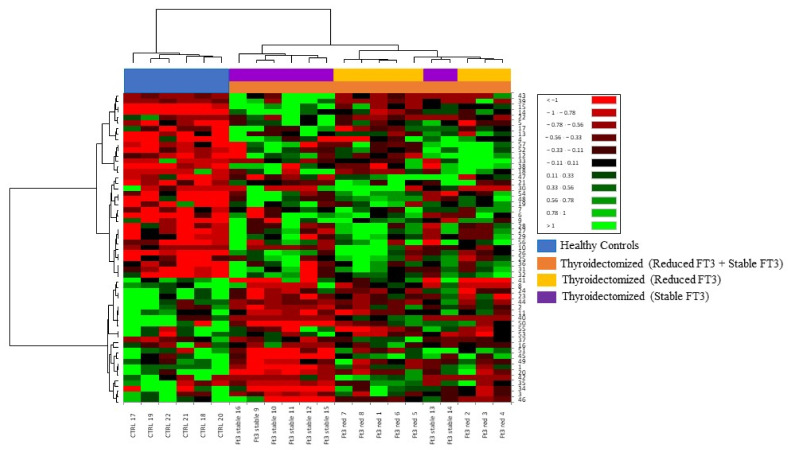
Heatmap branching of the 57 DASs separates samples into two principal clusters corresponding to thyroidectomized (orange bar) and control groups (blue bar). The thyroidectomized group is furtherly subdivided in two groups corresponding to reduced FT3 (yellow bar) and stable FT3 (purple bar). Numbers on the right of the heatmap correspond to the spot numbers of differential proteins reported in Table 3. Protein abundance in the heatmap goes from brilliant green (highly abundant) to red (lowly abundant) as shown on the scale reported on the right.

**Figure 6 jcm-11-01676-f006:**
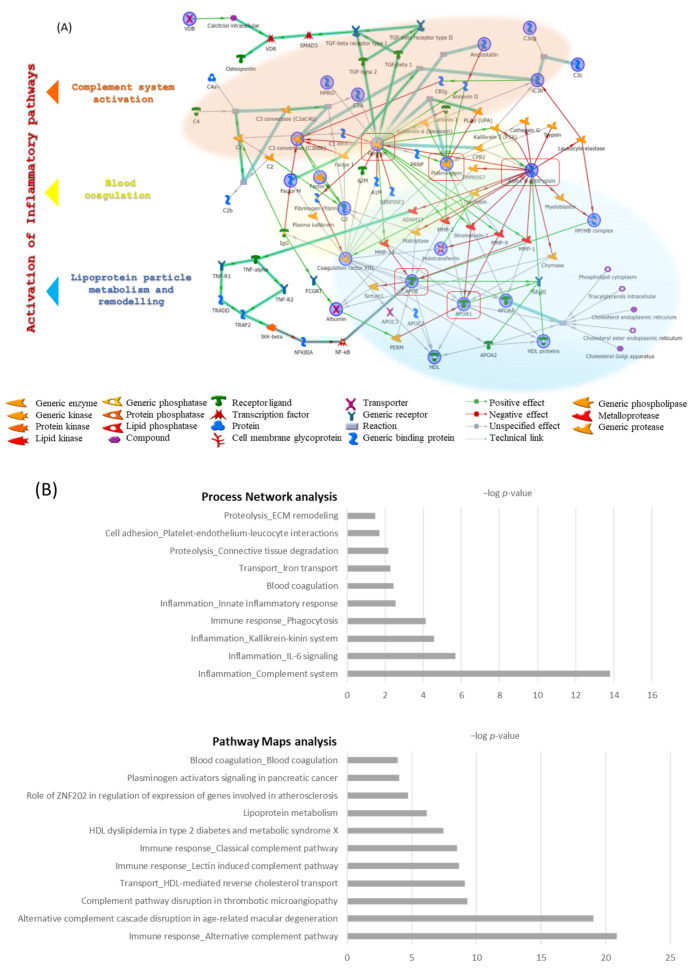
(**A**) Protein network built by MetaCore software, which uploaded the 34 differential abundant proteins (DAPs). Alpha 1-antitrypsin, apolipoprotein E (APOE), plasminogen, plasmin, and apolipoprotein AI (APOA1) are central functional hubs, circled in red. Three principal molecular paths are evidenced by colored clouds, respectively, in yellow corresponding to blood coagulation, orange as a complement system, and blue as lipoprotein particle remodeling. A legend of MetaCore symbols is reported under the protein network. (**B**) Histograms report the process network and pathway maps analyses results.

**Table 1 jcm-11-01676-t001:** Patients’ demographic data and comparison of post-surgical thyroid hormone levels between “reduced FT3” and “stable FT3” patients.

	Reduced FT3	Stable FT3	*p*-Value
Age (years)			0.2
Mean ± SD	46 ± 17.2	54 ± 7.4	
Range	24–73	44–65	
Median	44.5	54	
Sex			>0.9
Males (n, %)	4 (50%)	4 (50%)	
Females (n, %)	4(50%)	4 (50%)	
LT4 daily dose (mcg)			0.6
Mean ± SD	131.2 ± 28.4	125 ± 31.3	
Range	82–175	87.5–175	
Median	131.2	118.7	
Pro-Kg LT4 dose (mcg)			0.01
Mean ± SD	1.8 ± 0.3	1.5 ± 0.1	
Range	1.4–2.3	1.3–1.6	
Median	1.7	1.5	
Post-surgical FT3 (pg/mL)			0.02
Mean ± SD	2.8 ± 0.1	3.2 ± 0.3	
Range	2.6–3.2	2.7–3.6	
Median	2.9	3.2	
Normal range: 2.5–4.5			
Post-surgical FT4 (pg/mL)			0.4
Mean ± SD	11.2 ± 1.6	10.6 ± 1.3	
Range	8.3–13.6	8.5–12.3	
Median	11.6	10.4	
Normal range: 5.8–16.4			
Post-surgical TSH (mUI/L)			0.1
Mean ± SD			
Range	1 ± 0.5	0.6 ± 0.3	
Median	0.4–1.7	0.4–1.2	
Normal range: 0.4–4.0	0.98	0.6	
Post-surgical FT3/FT4			0.06
Mean ± SD	0.2 ± 0.04	0.3 ± 0.04	
Range	0.19–0.33	0.2–0.3	
Median	0.27	0.29	

**Table 2 jcm-11-01676-t002:** Pre- and post-surgical biochemical data according to the two subgroups: “stable FT3” and “reduced FT3”.

		Pre-Surgical	Post-Surgical	*p*-Value
“Stable FT3”	TSH (mUI/L)			0.6
	Mean ± SD	0.7 ± 0.3	0.6 ± 0.3	
	Range	0.4–1.6	0.4–1.2	
	Median	0.7	0.6	
	FT3 pg/mL			0.4
	Mean ± SD	3.3 ± 0.3	3.2 ± 0.3	
	Range	2.9–3.9	2.7–3.6	
	Median	3.3	3.2	
	FT4 pg/mL			0.06
	Mean ± SD	8.9 ± 1.1	10.6 ± 1.3	
	Range	7.3–10.7	8.5–12.3	
	Median	8.6	10.4	
“Reduced FT3”	TSH (mUI/L)			0.4
	Mean ± SD	0.9 ± 0.4	1 ± 0.5	
	Range	0.3–1.4	0.4–1.7	
	Median	0.8	0.98	
	FT3 pg/mL			<0.001
	Mean ± SD	3.7 ± 0.2	2.8 ± 0.1	
	Range	3.4–4.2	2.6–3.2	
	Median	3.7	2.9	
	FT4 pg/mL			0.9
	Mean ± SD	11.1 ± 3.3	11.2 ± 1.6	
	Range	7.7–17.7	8.3–13.6	
	Median	9.8	11.6	

**Table 3 jcm-11-01676-t003:** The table reports UniProtKB protein description, abbreviation, accession number (AC), theoretical isoelectric point (pI), and molecular weight (MW). Spot numbers correspond to those in Figure 1. Mascot search results were reported by score, number of matched peptides, and sequence coverage. Moreover, the table reports the %V means, Kruskal–Wallis *p*-values and %V mean protein ratio in stable FT3, reduced (Red) FT3, and controls (CTRL). Numbers in bold in “Ratio” column highlight the valid fold change ≥1.5.

Spot *n*	Protein Name	UniProt Entry (HUMAN)	AC	Theoretical pI/Mr (kDa)	Mascot Search Results in Swissprot db			Means			Kruscal Wallis *p*-Value	Ratio		
					Score	Matched Peptides	Coverage	Ft3red	Ft3	CTRL		Red/Stable	Red/CTRL	Stable/CTRL
									Stable					
**2**	**Complement Factor H**	CFAH	P08603	6.21/143,680	158	17	15%	0.0254	0.014	0.050	0.012	**1.836**	**−1.981**	**−3.639**
**4**	**Plasminogen**	PLMN	P00747	7.04/93,247	231	23	31%	0.0095	0.015	0.002	0.022	**−1.61**	**5.22**	**8.427**
**5**	**Plasminogen**	PLMN	P00747	7.04/93,247	264	22	28%	0.0135	0.026	0.005	0.004	**−1.97**	**2.736**	**5.389**
**6**	Complement factor B	CFAB	P00751	6.67/86,847	83	7	13%	0.0592	0.052	0.015	0.01	**1.141**	**4.001**	**3.508**
**7**	**Complement factor B**	CFAB	P00751	6.67/86,847	233	18	29%	0.0931	0.059	0.031	0.008	**1.575**	**2.974**	**1.888**
**9**	Serotransferrin	TRFE	P02787	6.81/79,294	96	9	12%	0.0297	0.025	0.007	0.028	**1.181**	**4.378**	**3.706**
**10**	**Albumin + Coagulation factor XIII B chain**	ALBU F13B	P02768 P05160	5.92/71,317 6.01/77,742	236 = 146 + 111	12 15	22% 22%	0.0244	0.007	0.002	0.033	**3.437**	**12.337**	**3.589**
**11**	Albumin + alpha-1B-glycoprotein	ALBU A1BG	P02768 P04217	5.92/71,317 5.56/54,790	269 = 163 + 133	19 15	34% 42%	0.0706	0.066	0.147	0.024	1.072	**−2.09**	**−2.24**
**15**	**Complement factor B frag. C-term**	CFAB	P00751	6.67/86,847	104	11	17%	0.0215	0.041	0.002	0.0001	**−1.91**	**11.445**	**21.831**
**16**	**Vitamin D-binding protein + Alpha-1-antitrypsin**	VTDB A1AT	P02774 P01009	5.32/54,480 5.37/46,878	375 = 230 + 158	21 17	43% 39%	0.0506	0.028	0.083	0.04	**1.822**	**−1.634**	**−2.978**
**17**	**Alpha-1-antitrypsin + Apolipoprotein A-I**	A1AT APOA1	P01009 P02647	5.37/46,878 5.56/30,759	437 = 295 + 150	29 15	70% 49%	0.1443	0.236	0.112	0.01	**−1.63**	1.292	**2.111**
**19**	Albumin fragment N-term	ALBU	P02768	5.92/71,317	244	23	38%	0.0445	0.044	0.022	0.036	1.013	**2.013**	**1.986**
**20**	**Apolipoprotein A-IV + Haptoglobin**	APOA4 HPT	P06727 P00738	5.528/45,344 6.13/45,861	443 = 352 + 116	31 14	58% 29%	0.0755	0.035	0.206	0.003	**2.131**	**−2.73**	**−5.82**
**21**	Albumin fragment C-term + Haptoglobin	ALBU HPT	P02768 P00738	5.92/71,317 6.13/45,861	235 = 139 + 119	16 14	27% 36%	0.0496	0.037	0.016	0.007	1.327	**3.194**	**2.408**
**22**	**Histidine-rich glycoprotein**	HRG	P04196	7.09/60,510	179	14	26%	0.004	0.021	0.007	0.03	**−5.17**	**−1.788**	**2.893**
**23**	Complement C3 fragment C-term + Histidine-rich glycoprotein	CO3 HRG	P01024 P04196	6.02/188,569 7.09/60,510	280 = 166 + 114	28 14	18% 29%	0.056	0.049	0.140	0.007	1.152	**−2.501**	**−2.88**
**26**	Albumin fragment N-term	ALBU	P02768	5.92/71,317	253	20	26%	0.047	0.051	0.0033	0.002	0.924	**14.432**	**15.622**
**30**	**Complement C1q subcomponent subunit B**	C1QB	P02746	8.83/26,933	157	12	44%	0.0114	0.007	0.005	0.03	**1.676**	**2.283**	1.363
**34**	**Apolipoprotein A-I**	APOA1	P02647	5.56/30,759	334	28	70%	0.0832	0.031	0.116	0.02	**2.674**	0.717	**−3.728**
**35**	**Apolipoprotein A-I**	APOA1	P02647	5.56/30,759	321	26	70%	0.0964	0.025	0.1020	0.05	**3.851**	0.944	**−4.079**
**37**	**Hemoglobin**	HBA	P69905	8.72/15,305	128	7	47%	0.0985	0.04	0.044	0.04	**2.757**	**2.257**	0.819
**39**	**Apolipoprotein A-I fragment C-term**	APOA1	P02647	5.56/30,759	100	10	34%	0.0141	0.04	0.0019	0.03	**−2.83**	**7.398**	**20.97**
**40**	**Complement C1s subcomponent**	C1S	P09871	4.86/78,174	250	25	32%	0.0067	0	0.0188	0.02	**0.007**	**−2.798**	**0/0.018**
**41**	Complement C1s subcomponent	C1S	P09871	4.86/78,174	91	7	16%	0.0112	0.012	0.0302	0.03	0.946	**−2.701**	**−2.556**
**42**	**Complement C1s subcomponent**	C1S	P09871	4.86/78,174	150	14	28%	0.007	0.001	0.009	0.02	**7.793**	0.746	**−10.447**
**43**	**Apolipoprotein A-I + Alpha-1-antitrypsin**	APOA1 A1AT	P02647 P01009	5.56/30,759 5.37/46,878	172 = 109 + 96	10 10	35% 23%	0.0172	0.083	0.005	0.03	**−4.86**	**3.156**	**15.324**
**44**	Apolipoprotein A-I fragment N-term	APOA1	P02647	5.56/30,759	199	16	41%	0.0127	0.019	0.0914	0.01	0.661	**−7.204**	**−4.759**
**45**	**Alpha-1-antitrypsin + Apolipoprotein A-I**	A1AT APOA1	P01009 P02647	5.37/46,878 5.56/30,759	264 = 165 + 112	16 11	34% 38%	0.7654	0.306	0.777	0.009	**2.502**	0.986	**−2.539**
**47**	Apolipoprotein E	APOE	P02649	5.44/122,983	325	26	67%	0.2241	0.189	0.081	0.03	1.185	**2.777**	**2.343**
**50**	**Apolipoprotein A-I + Alpha-1-antitrypsin**	APOA1 A1AT	P02647 P01009	5.56/30,759 5.37/46,878	329 = 202 + 123	19 14	58% 34%	0.6651	0.367	1.157	0.02	**1.812**	**−1.74**	**−3.154**
**51**	Complement Factor H	CFAH	P08603	6.21/143,680	269	25	23%	0.0383	0.026	0.047	0.03	1.495	0.824	**−1.814**
**55**	Alpha-1B-glycoprotein	A1BG	P04217	5.56/54,790	207	16	36%	0.0756	0.109	0.137	0.023	0.692	**−1.81**	0.796
**56**	Albumin fragment C-term	ALBU	P02768	5.92/71,317	253	22	37%	0.0819	0.071	0.040	0.012	1.16	**2.033**	**1.75**
**57**	Immunoglobulin heavy constant gamma 2	IGHG2	P01859	7.66/36,505	54	16	46%	0.082	0.074	0.049	0.03	1.09	**1.666**	**1.519**

## Data Availability

The mass spectrometry proteomics data have been deposited to the ProteomeXchange Consortium via the PRIDE [29] partner repository with the dataset identifier PXD028378.

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
