# Peer review of "Alteration of Serum Proteome in Levo-Thyroxine-Euthyroid Thyroidectomized Patients"

_jcm, 2022, doi:10.3390/jcm11061676_

Round 1
Reviewer 1 Report
It has been suggested that LT4 supplementation may not ensure tissue euthyroidism in all hypothyroid patients, which can be a possible explanation for residual symptoms in these patients. Lower FT3 serum levels may potentially reflect lower thyroid hormone levels in peripheral tissues. In the current study, Landi et al. investigated the differences in serum proteomic profiles between thyroidectomized patients on LT4 supplementation with "reduced" and "unchanged" FT3 levels, as compared to the individual pre-surgical FT3 levels. The authors report significant differences in serum proteome between these two groups, especially in A1AT and APOAI profiles. Further enrichment analyses suggest that differential proteins are involved in the complement system activation, coagulation and lipoprotein metabolism, which altogether may promote the pro-inflammatory response. These are novel and valuable findings that require further replication.
I have the following comments:
1. The reviewer highly appreciates the idea of this study, as the problem of residual complaints in hypothyroid patients on LT4 treatment is very important from a clinical perspective and the underlying mechanisms for this phenomenon are still unknown.
2. Given the multi-level analyses based on different approaches used in the study, the manuscript would benefit from a flowchart illustrating each stage of the analysis process.
3. Why did the authors decide to use Kruskal–Wallis test that compares three groups of subjects ("reduced" FT3, "unchanged" FT3 and healthy controls) instead of performing a direct comparison of protein profile between "reduced" FT3 and "unchanged" FT3 patients or "reduced" FT3 patients and healthy controls? It is currently unclear if the observed statistically significant differences reflect different protein profiles in "reduced" FT3 and "unchanged" FT3 patients or rather thyroidectomized patients and healthy controls.
4. Figures and tables should be self-expentatory – please provide more robust legends and abbreviations lists for each figure and table. Please especially provide an explanation for the Heatmap caption.
5. The manuscript would benefit from some style editing and interpunction correction: currently some long sentences with improper interpunction are very difficult to follow or even unclear.
6. Since "This research received no external funding" and "Authors have no competing interest to declare", please clarify how "MERCK contributed to the work".
7. Please clarify why ethical review and approval was waived for this study.
Author Response
Siena, 2nd March 2022
RE: Alteration of serum proteome in levo-thyroxine-euthyroid thyroidectomized patients by Landi C. et al.
To the Editor,
The authors would like to thank you and the reviewers for your precious comments about our manuscript. We submit a revised version of the above work. The text was modified according to the referees’ helpful suggestions.
A new author, Dr Alessio Bombardieri, has been added.
Our point-by-point replies appear below. All the modifications in the manuscript are reported in red in the track changes version.
Reviewer 1
It has been suggested that LT4 supplementation may not ensure tissue euthyroidism in all hypothyroid patients, which can be a possible explanation for residual symptoms in these patients. Lower FT3 serum levels may potentially reflect lower thyroid hormone levels in peripheral tissues. In the current study, Landi et al. investigated the differences in serum proteomic profiles between thyroidectomized patients on LT4 supplementation with "reduced" and "unchanged" FT3 levels, as compared to the individual pre-surgical FT3 levels. The authors report significant differences in serum proteome between these two groups, especially in A1AT and APOAI profiles. Further enrichment analyses suggest that differential proteins are involved in the complement system activation, coagulation and lipoprotein metabolism, which altogether may promote the pro-inflammatory response. These are novel and valuable findings that require further replication.
I have the following comments:
- The reviewer highly appreciates the idea of this study, as the problem of residual complaints in hypothyroid patients on LT4 treatment is very important from a clinical perspective and the underlying mechanisms for this phenomenon are still unknown.
A1. We really thank the reviewer for the observations and the appreciation regarding the importance of this research.
- Given the multi-level analyses based on different approaches used in the study, the manuscript would benefit from a flowchart illustrating each stage of the analysis process.
A2. According to the helpful suggestion of the reviewer, we introduce a graphical abstract reporting the analysis process.
- Why did the authors decide to use Kruskal–Wallis test that compares three groups of subjects ("reduced" FT3, "unchanged" FT3 and healthy controls) instead of performing a direct comparison of protein profile between "reduced" FT3 and "unchanged" FT3 patients or "reduced" FT3 patients and healthy controls? It is currently unclear if the observed statistically significant differences reflect different protein profiles in "reduced" FT3 and "unchanged" FT3 patients or rather thyroidectomized patients and healthy controls.
A3. We thank the reviewer for the interesting question. We decide to perform the Kruskal-Wallis test because we consider the behaviour of all spots in all samples in three conditions and since the nature of the sample, our data have not a normal distribution. The observed statistically significant differences reflect homogeneously the spots behaviour in controls, reduced FT3 and stable FT3 samples and the abundance tendency can be observed on the heatmap analysis graph. However, to better visualize the differential proteome between reduced and stable FT3 sera, specific statistically differential proteins are reported in bold in table 3 on the basis of the Red/stable ratio.
- Figures and tables should be self-expentatory – please provide more robust legends and abbreviations lists for each figure and table. Please especially provide an explanation for the Heatmap caption.
A4. We provide more robust legends for tables and figures. Figure 5 has been modified in order to improve the understanding of heatmap results. We have also introduced the MetaCore legend to better understand symbols reported in the protein network. We hope that table and figure captions now are in line with the reviewer requests.
- The manuscript would benefit from some style editing and interpunction correction: currently some long sentences with improper interpunction are very difficult to follow or even unclear.
A5. We thank the reviewer for the suggestion. We review and modify some sentences along the text.
- Since "This research received no external funding" and "Authors have no competing interest to declare", please clarify how "MERCK contributed to the work".
A6. We thank MERCK since one of the authors has a MERCK research fellowship, but no external fundings were received for consumables or other expenses directly related to the research.
- Please clarify why ethical review and approval was waived for this study.
A7. This study has been approved by our local ethical committee (#18923) and all subjects agree to be enrolled in the research. We missed in the first version the number of protocol approval and this was maybe misleading. We have added this information in the revised version.
We trust the paper is now in order for acceptance.
Yours sincerely,
Claudia Landi, Maria Grazia Castagna, Silvia Cantara and colleagues
Siena University, Italy

Reviewer 2 Report
This is a very very interesting paper that addresses a small but important subset of patients post thyroidectomy. I feel that the number of participants in this study limits the valuable work performed and as such I think there needs to be at least a power analysis before this can be published
Author Response
Siena, 2nd March 2022
RE: Alteration of serum proteome in levo-thyroxine-euthyroid thyroidectomized patients by Landi C. et al.
To the Editor,
The authors would like to thank you and the reviewers for your precious comments about our manuscript. We submit a revised version of the above work. The text was modified according to the referees’ helpful suggestions.
A new author, Dr Alessio Bombardieri, has been added.
Our point-by-point replies appear below. All the modifications in the manuscript are reported in red in the track changes version.
Reviewer 2
This is a very very interesting paper that addresses a small but important subset of patients post thyroidectomy. I feel that the number of participants in this study limits the valuable work performed and as such I think there needs to be at least a power analysis before this can be published.
We thank the reviewer for the appreciation regarding the importance of this research and we thank for the important suggestion.
Due to our dataset accounted thousand variables, we mainly focused on avoiding type 1 errors through the postHoc correction. However, we opted for the Benjamini-Hochberg FDR rather than other corrections (like Bonferroni) because BH-FDR resulted in lower rate of the type 2 errors.
Albeit we might assume an higher Cohen effect size considering the overal high correlation of our variables, the power value of our analysis would be not appropriate; considering a sample size of 8, a significativity level of 0.05 and a Cohen effect size of 0.75, the power would be close to 0.3. The recommended sample size to have a decent power (power > 0.8) would require 30 cases but we could not have the opportunity to recruit this amount of patients.
The size of this population is not considered small for a preliminary proteomic study since proteomic is time and cost consuming. Moreover, the incidence of hypothyroidism symptoms in levo-thyroxine-euthyroid thyroidectomized patients is considered not so frequent. For these reasons this population of samples could be considered appropriate for a preliminary proteomic analysis. We have stressed that this is a preliminary study both in the abstract and in the introduction.
We trust the paper is now in order for acceptance.
Yours sincerely,
Claudia Landi, Maria Grazia Castagna, Silvia Cantara and colleagues
Siena University, Italy
Reviewer 3 Report
To evaluate the FT3 levels in athyreotic patients who taking LT4 therapy this author made case-control study. From this study this author concluded that patients undergoing thyroidectomy are normally treated with LT4 replacement therapy but a substantial proportion of them experience hypothyroid-like symptoms despite normal TSH levels. I thought this study contains many critical issues that should be solved. I commented those issues as following.
Over all
The main findings of the present study are that patients undergoing thyroidectomy are normally treated with LT4 replacement therapy but a substantial proportion of them experience hypothyroid-like condition despite normal TSH levels.
- Then the secretory capacity of the pituitary gland (Jostetl’s TSH index) should be calculated. Even by proteomic analysis and by enrichment analysis, the patients undergoing thyroidectomy with taking LT4 replacement therapy shows similar condition for overt hypothyroid, this dose not cover the secretory capacity of the pituitary gland (Jostetl’s TSH index).
- Furthermore, this author also focused on the peripheral conversion of T4 to T3. Then the influence of global step-up deiodinase activity, maximum thyroid secretory capacity (SPINA-GT), and the overall activity of peripheral 5¢-deiodinases (SPINA-GD) also should be taken into consideration.
About title
- In title, LT4 should be spell out.
About abstract
- There is no information about the size of study sample and study method. Then redear could not image what type of study this author made and how large the sample of present study that this author performed.
- Even this is an original article, no specific analysis result was shown.
- Hypothesis of present study should be shown in abstract. In present description, reader of present manuscript could not understand what this author wants to clarified.
- This author found the similar condition for athyreotic patients who taking LT4 therapy to overt hypothyroidism. However, no description that showed those specific results. Then readers could not image what variable shows similar value between those participants. And how similar those variables were.
In introduction section
- This sutho described as “patients undergoing thyroidectomy need to maintain correct thyroid hormone (TH) levels by replacement therapy with levo-thyroxine (LT4).” What is the thyroid hormone mean here? Activated thyroid hormone level ? FT3/FT4 ratio ? Or FT4 levels?
- This author described as “we hypothesized that the lower plasmatic FT3 levels observed in a subgroup of athyreotic patients treated with LT4 therapy, could be associated with tissue hypothyroidism. However, I could not understand the scientific value of this hypothesis. From this description, I thought this author wants to clarified that inappropriate therapy causes inappropriate results. Then this study showed that deficient of thyroid hormone replacement therapy causes similar symptom to hypothyroidism. I thought this is so basic and this manuscript could not holds enough newly information to be published.
- Still there is no description what type of study this author made. And there is also no description about the size of present study.
Method section
- References that showed the normal range of FT3, FT4 and TSH should be shown.
- This author described as “We arbitrarily selected a change of at least 0.5 pg/ml as a significant variation between pre- and post-surgical FT3 value. According to this criterion, patients were classified as having “reduced FT3” when post-surgical FT3 levels were at least 0.5 pg/ml lower than pre-surgical FT3 values”. However, these is no description that validate the using 0.5pg/mL as a cut off point for present definition. This is mandatory, because present study is a case-controlled study with small number of sample.
- Even there is no description about the control group in method section, this author used the sample of 6 healthy control. The method to choses those 6 healthy controls should be clarified. In addition to that, the reason why this author thought 6 is enough number for control in present study population also should be clarified.
- Analysis among participants with hypothyroidism also necessary in present study.
- How is the reproducibility of present analysis.
Discussion
- Clinical implication for present study should be described because the journal this author submitted was “Journal of Clinical Medicine”. Measuring FT3, FT4, TSH, and calculating Jostetl’s TSH index is much easy way to evaluate thyroid function than making proteomic analysis in daily clinical practice. Therefore, utility of proteomic analysis and enrichment analysis in clinical practice should be discussed.
- The description should be emphasized on the difference among the results of healthy control, hypothyroid control, thyroidectomized patient with FT3 stable and thyroidectomized FT3 reduced.
- This author concluded that LT4 replacement therapy might not restore euthyroid conditions in all thyroidectomized patients. This conclusion rise the question about how the influence of LT4 replacement therapy on participants with hypothyroidism who were not taking thyroidectomized therapy.
Author Response
Siena, 2nd March 2022
RE: Alteration of serum proteome in levo-thyroxine-euthyroid thyroidectomized patients by Landi C. et al.
To the Editor,
The authors would like to thank you and the reviewers for your precious comments about our manuscript. We submit a revised version of the above work. The text was modified according to the referees’ helpful suggestions.
A new author, Dr Alessio Bombardieri, has been added.
Our point-by-point replies appear below. All the modifications in the manuscript are reported in red in the track changes version.
Reviewer 3
To evaluate the FT3 levels in athyreotic patients who taking LT4 therapy this author made case-control study. From this study this author concluded that patients undergoing thyroidectomy are normally treated with LT4 replacement therapy but a substantial proportion of them experience hypothyroid-like symptoms despite normal TSH levels. I thought this study contains many critical issues that should be solved. I commented those issues as following.
We really appreciate the reviewer for the scrupulous reading of the paper and for the numerous food for thought to be considered also for further analyses. We try to answer to the comments in order to increase the quality of the manuscript.
Over all
The main findings of the present study are that patients undergoing thyroidectomy are normally treated with LT4 replacement therapy but a substantial proportion of them experience hypothyroid-like condition despite normal TSH levels.
- Then the secretory capacity of the pituitary gland (Jostetl’s TSH index) should be calculated. Even by proteomic analysis and by enrichment analysis, the patients undergoing thyroidectomy with taking LT4 replacement therapy shows similar condition for overt hypothyroid, this dose not cover the secretory capacity of the pituitary gland (Jostetl’s TSH index).
A1. The main finding of this paper is not that patients undergoing thyroidectomy are treated with LT4 and neither that some of them experience hypothyroid-like symptoms despite normal TSH levels (this is a clinical evidence). The main finding, highlighted by proteomic and enrichment analysis, is that those patients with reduced FT3 have a different proteome profile respect to "unchanged" FT3 patients.
However, as suggested, we have calculated pre and post surgical TSH index (TSHI) using the software SPINA Thyr 4.2.0 (version for windows). For all patients included in the study both pre and post surgical TSHI were between normal range indicating a normal secretory capacity of the pituitary gland and efficient feedback both by endogenous T4 and by the replacement therapy with LT4.
- Furthermore, this author also focused on the peripheral conversion of T4 to T3. Then the influence of global step-up deiodinase activity, maximum thyroid secretory capacity (SPINA-GT), and the overall activity of peripheral 5¢-deiodinases (SPINA-GD) also should be taken into consideration.
A2. In this article we do not focus on desiodase activity. We have already showed in a previous study that a polymorphism in DIO2 may lower DIO2 activity (reference #5), but not all patients complaining with post surgical hypothyroid symptoms carry this variant. This is the reason why, in this study, we search for other mechanisms possibly related to this clinical evidence.
Anyway, the SPINA Thyr 4.2.0 version allows to calculate also SPINA-GT and SPINA-GD. We then performed the analysis for all patients. In the post surgical, SPINA-GT was not available as these patients are on LT4 therapy. In the presurgical, we have a maximum thyroid secretory capacity (SPINA-GT) ranging from 4.19 to 7.46 pmol/s. This activity can be considered normal (reference range 1,4-8,7 pmol/s). Regarding the overall desiodases activity (SPINA-GD), this was in the reference range (20-40 noml/s) for all patients after surgery (21.87-37.75 nmol/s) and increased (maximum 59.77 nmol/s) in 4 patients. Of these, 2 had a reduced postsurgical FT3 and 2 had an "unchanged" postsurgical FT3.
Considering all indices together, patients seem to have normal pituitary and thyroid activity as well as a functioning negative feedback (TSHI) and desiodase activity (considered all together without distinction between DIO2 and other forms) strengthening the idea that other mechanisms may be implicated with post surgical persistent hypothyroidism.
About title
- In title, LT4 should be spell out.
A3. In the title LT4 has been spell out
About abstract
- There is no information about the size of study sample and study method. Then reader could not image what type of study this author made and how large the sample of present study that this author performed.
- Even this is an original article, no specific analysis result was shown.
- Hypothesis of present study should be shown in abstract. In present description, reader of present manuscript could not understand what this author wants to clarified.
A4, A5, A6. We thank the reviewer for the helpful suggestions. In order to improve the description of our research we rewrite the abstract as follow:
“The monotherapy with levo-thyroxine (LT4) is the treatment of choice for patients with hypo-thyroidism after thyroidectomy. However, many athyreotic LT4-treated patients with thyroid hormones in the physiological range experience hypothyroid-like symptoms showing post-operative statistically significant lower FT3 levels with respect to that before total thyroidectomy. Since we hypothesized that the lower plasmatic FT3 levels observed in this subgroup, could be associated with tissue hypothyroidism, here we compared, by a preliminary proteomic analysis, 8 sera of patients with reduced post-surgical FT3 to 8 sera from patients with FT3 levels similar to pre-surgery levels, and 6 healthy controls. Proteomic analysis highlights a different serum protein profile among the considered conditions. By enrichment analysis, differential proteins are involved in coagulation processes (PLMN -1.61, -1.98 in Reduced vs stable FT3, p<0.02; A1AT fragmentation), complement system activation (CFAH +1.83, CFAB +1.5, C1Qb +1.6, C1S +7.79 in Reduced vs Stable FT3, p<0.01) and in lipoprotein particles remodeling (APOAI fragmentation; APOAIV +2.13, p<0.003), potentially leading to a pro-inflammatory response. This study suggests that LT4 replacement therapy might restore biochemical euthyroid conditions in thyroidectomized patients but in some cases without re-establish body tissues euthyroidism. Since our results, this condition is reflected by the serum protein profile.”
- This author found the similar condition for athyreotic patients who taking LT4 therapy to overt hypothyroidism. However, no description that showed those specific results. Then readers could not image what variable shows similar value between those participants. And how similar those variables were.
A7. We thank the reviewer for the observation. In our discussion we suggest a similarity among molecular pathways dysregulated in hypothyroidism with the molecular pathways that we found by MetaCore analysis relies on the differentially identified proteins that we found.
Proteomic shows an overview of the pathological molecular mechanisms that need to be further studied. Part of dysregulated proteins we found and their involvement in particular molecular pathways such as blood coagulation, complement system, and lipoprotein particles remodelling, in some cases have been already described as related to hypothyroidism as reported in other researches. We only comment these similarities in introduction and discussion sections.
Obviously, there is no a precise variable but there is an altered modulation of some players in the same molecular mechanisms. Actually, there is the necessity to validate the dysregulation of these metabolic pathways on this particular subgroup of patients, but the numerous affinities with previous works on hypothyroid patients suggest a discussion regarding the similarity among these conditions.
- Pagni, F.; L’Imperio, V.; Bono, F.; Garancini, M.; Roversi, G.; De Sio, G.; Galli, M.; Smith, A.J.; Chinello, C.; Magni, F. Proteome Analysis in Thyroid Pathology. Expert Rev Proteomics 2015, 12, 375–390, doi:10.1586/14789450.2015.1062369.
- Engelmann, B.; Bischof, J.; Dirk, A.-L.; Friedrich, N.; Hammer, E.; Thiele, T.; Führer, D.; Homuth, G.; Brabant, G.; Völker, U. Effect of Experimental Thyrotoxicosis onto Blood Coagulation: A Proteomics Study. Eur Thyroid J 2015, 4, 119–124, doi:10.1159/000381769.
- M, P.; B, E.; T, K.; J, G.; Al, D.; E, H.; Ka, I.; M, N.; H, W.; D, F.; et al. Plasma Proteome and Metabolome Characterization of an Experimental Human Thyrotoxicosis Model Available online: https://pubmed.ncbi.nlm.nih.gov/28065164/ (accessed on 2 December 2020).
- Alfadda, A.A.; Benabdelkamel, H.; Masood, A.; Jammah, A.A.; Ekhzaimy, A.A. Differences in the Plasma Proteome of Patients with Hypothyroidism before and after Thyroid Hormone Replacement: A Proteomic Analysis. Int J Mol Sci 2018, 19, doi:10.3390/ijms19010088.
- Bitencourt, C.S.; Duarte, C.G.; Azzolini, A.E.C.S.; Assis-Pandochi, A.I. Alternative Complement Pathway and Factor B Activities in Rats with Altered Blood Levels of Thyroid Hormone. Braz J Med Biol Res 2012, 45, 216–221, doi:10.1590/s0100-879x2012007500028.
- Türemen, E.E.; Çetinarslan, B.; Åžahin, T.; Cantürk, Z.; Tarkun, Ä°. Endothelial Dysfunction and Low Grade Chronic Inflammation in Subclinical Hypothyroidism Due to Autoimmune Thyroiditis. Endocr J 2011, 58, 349–354, doi:10.1507/endocrj.k10e-333.
- Chadarevian, R.; Bruckert, E.; Leenhardt, L.; Giral, P.; Ankri, A.; Turpin, G. Components of the Fibrinolytic System Are Differently Altered in Moderate and Severe Hypothyroidism. J Clin Endocrinol Metab 2001, 86, 732–737, doi:10.1210/jcem.86.2.7221
- Jung, K.Y.; Ahn, H.Y.; Han, S.K.; Park, Y.J.; Cho, B.Y.; Moon, M.K. Association between Thyroid Function and Lipid Profiles, Apolipoproteins, and High-Density Lipoprotein Function. J Clin Lipidol 2017, 11, 1347–1353, doi:10.1016/j.jacl.2017.08.015.
- O’Brien, T.; Katz, K.; Hodge, D.; Nguyen, T.T.; Kottke, B.A.; Hay, I.D. The Effect of the Treatment of Hypothyroidism and Hyperthyroidism on Plasma Lipids and Apolipoproteins AI, AII and E. Clin Endocrinol (Oxf) 1997, 46, 17–20, doi:10.1046/j.1365-2265.1997.d01-1753.x.
- Zhou, J.; Cheng, G.; Pang, H.; Liu, Q.; Liu, Y. The Effect of 131I-Induced Hypothyroidism on the Levels of Nitric Oxide (NO), Interleukin 6 (IL-6), Tumor Necrosis Factor Alpha (TNF-α), Total Nitric Oxide Synthase (NOS) Activity, and Expression of NOS Isoforms in Rats. Bosn J Basic Med Sci 2018, 18, 305–312, doi:10.17305/bjbms.2018.2350.
- Mancini, A.; Di Segni, C.; Raimondo, S.; Olivieri, G.; Silvestrini, A.; Meucci, E.; Currò, D. Thyroid Hormones, Oxidative Stress, and Inflammation. Mediators Inflamm 2016, 2016, 6757154, doi:10.1155/2016/6757154.
In introduction section
- This sutho described as “patients undergoing thyroidectomy need to maintain correct thyroid hormone (TH) levels by replacement therapy with levo-thyroxine (LT4).” What is the thyroid hormone mean here? Activated thyroid hormone level ? FT3/FT4 ratio ? Or FT4 levels?
A8. Accordingly to current guide lines, TSH and Free T4 are routinely evaluated to verify if athyreotic patients are adequately treated with levo-thyroxine therapy (Ref. #1).
- This author described as “we hypothesized that the lower plasmatic FT3 levels observed in a subgroup of athyreotic patients treated with LT4 therapy, could be associated with tissue hypothyroidism. However, I could not understand the scientific value of this hypothesis. From this description, I thought this author wants to clarified that inappropriate therapy causes inappropriate results. Then this study showed that deficient of thyroid hormone replacement therapy causes similar symptom to hypothyroidism. I thought this is so basic and this manuscript could not holds enough newly information to be published.
A9. Probability we were not clear in explaining the aim of the study nor the inclusion criteria we used. We try to clarify these issues. We are, of course, aware that inappropriate therapy may not compensate thyroid hormones correctly. Unfortunately, to date, the main challenge in the management of patients with hypothyroidism is the persistence of hypothyroid symptoms in a subgroup of patients treated with LT4 despite normal TSH and FT4 levels.
Indeed, patients included in this study were adequately treated (euthyroid patients during LT4 therapy) as confirmed by normal to TSH and FT4 levels.
Our patients had lower FT3 (but still in the normal range) only when compared with FT3 levels obtained, in the same patients, before surgery. We aimed to verify, if these lower FT3 levels (compared with pre-surgical FT3) might be associated with different proteomic profiles. Our results showed that this small decrease of FT3 was sufficient to change proteomic profile at tissutal levels and, probably, it may be responsible of the symptoms complain by patients. However additional studies, including a larger cohort of patients will be necessary to confirm our hypothesis.
- Still there is no description what type of study this author made. And there is also no description about the size of present study.
A10. We very thank the reviewer for the suggestion. To this purpose we add into the introduction the following sentence to page 2, line 63:
“To explore this hypothesis, here we perform a preliminary proteomic analysis on a selected population of 8 sera from patients with reduced post-surgery FT3 and 8 sera from patients with stable post-surgery FT3 levels. Moreover, sera from 6 subjects without hypothyroidism problems and without comorbidities were considered as healthy control group for the comparison. Dysregulated protein spots, found by proteomic, were considered for principal component and heatmap analyses. Identified proteins were submitted to enrichment analysis in order to highlight their position into specific molecular pathways. Interesting proteins and their behavior were validated by two dimensional western blot in an alternative cohort of samples to confirm their particular pattern and their amount in reduced FT3.”
Method section
- References that showed the normal range of FT3, FT4 and TSH should be shown.
A11.We perform our test by Access Immunoassay Systems 2006, Beckman Coulter. The manufacturer indicates for each test reference range, limit of detection, sensitivity and sensibility. We refer to the range indicated by Beckman.
- This author described as “We arbitrarily selected a change of at least 0.5 pg/ml as a significant variation between pre- and post-surgical FT3 value. According to this criterion, patients were classified as having “reduced FT3” when post-surgical FT3 levels were at least 0.5 pg/ml lower than pre-surgical FT3 values”. However, these is no description that validate the using 0.5pg/mL as a cut off point for present definition. This is mandatory, because present study is a case-controlled study with small number of sample.
A12. As reported in our previous paper (Ref. # 5), we define “reduced FT3” levels when the post-surgical FT3 levels was at least lower than 0.5 pg/ml respect to pre-surgical FT3 values. In our laboratory, the FT3 interassay CV is 8%. The difference of 0.5 pg/ml that we decided to be significant, was superior that 8% CV for each patient, suggesting that the reduction was clinically relevant.
- Even there is no description about the control group in method section, this author used the sample of 6 healthy control. The method to choses those 6 healthy controls should be clarified. In addition to that, the reason why this author thought 6 is enough number for control in present study population also should be clarified.
A13. We thank the reviewer for the observation. In Material and Method section we add the following sentence to describe the control group to page 3, line 98:
“Control subjects were matched for age and gender. They had no history of concomitant pathologies and were not on any therapy.”
We add 6 healthy subjects present in our databank that matched for age and gender with no history of concomitant pathologies and not on any therapy in order to have as homogeneous as possible group with the other conditions.
- Analysis among participants with hypothyroidism also necessary in present study.
A14. We thank the reviewer for raising this point. We choose to focalize our proteomic analysis on this peculiar pathological situation because, to the best of our knowledge, there is no information about the cause of hypothyroidism symptoms referred by these patients, except for thyroid hormone levels that, for the current knowledge, not completely explain these symptoms and the reduced quality of life. Moreover, we did not take into consideration hypothyroidism in this analysis because there are proteomic studies already done on difference between healthy controls and hypothyroidism. Their results are compared with ours in the introduction and discussion sections as reported above (A7).
- How is the reproducibility of present analysis.
A15. Proteomic analysis performed in this work is a preliminary analysis. Normally proteomics is not performed on higher number of samples and in this case is acceptable due to the low prevalence of athyreotic LT4-treated patients with hypothyroid-like symptoms showing post-operative statistically significant lower FT3 levels with respect to that before total thyroidectomy. The contemporary visualization of more dysregulated proteins permits to have an overview on the pathological mechanisms suggesting further hypothesis to be validated. The reproducibility value is in the validation step downstream to the proteomic analysis. However, we start this analysis considering only 4 patients per condition and when we increased the population to 8 patients, results didn’t change. The principal involved pathways were always the same. Obviously, this kind of analysis is preliminary but already suggest interesting biomarkers and their involvement in specific molecular pathways. For instance, APOAI and A1AT amount and fragmentation have been validated in this study by two-dimensional western blot reporting the same behaviour observed by proteomic and mass spectrometry. Moreover, in order to evaluate the modulation of the reported molecular pathways, clinical analyses could be done on these patients regarding blood coagulation levels, lipoparticles remodelling, complement activation and levels of inflammation.
Discussion
- Clinical implication for present study should be described because the journal this author submitted was “Journal of Clinical Medicine”. Measuring FT3, FT4, TSH, and calculating Jostetl’s TSH index is much easy way to evaluate thyroid function than making proteomic analysis in daily clinical practice. Therefore, utility of proteomic analysis and enrichment analysis in clinical practice should be discussed.
A16. We thank the reviewer for the observation. Obviously proteomic is not used for clinical practice since it is time and cost consuming. Proteomics is used for research purposes, to highlight potential biomarkers that after validations could be used in clinical practice (translational medicine). At the same time blood coagulation levels, complement activation, lipoprotein particle composition and levels of inflammation are suggestions that could be evaluated in these patients to considerate their modulation in euthyroid patients that experienced symptoms of hypothyroidism.
To improve the discussion, we add at page 13, line 282 the sentence:
“Proteomics is used for research purposes, to highlight potential biomarkers that could be used in clinical practice (translational medicine).”
And
To page 15, line 368 we add:
“Moreover, to the clinical point of view, blood coagulation levels, complement activation, lipoprotein particle composition and levels of inflammation could be evaluated in levo-thyroxine-euthyroid thyroidectomized patients to considerate their modulation.”
- The description should be emphasized on the difference among the results of healthy control, hypothyroid control, thyroidectomized patient with FT3 stable and thyroidectomized FT3 reduced.
A17. We thank the reviewer for this suggestion. We did not report results regarding healthy controls vs hypothyroid control patients because are already present in other proteomic studies, for this reason, we focalize our study only on levo-thyroxine-euthyroid thyroidectomized patients with stable, reduced FT3 and healthy controls.
However, to emphasize the difference among FT3 stable and thyroidectomized FT3 reduced, we made some modifications:
Table 3 was modified reporting in bold the differential proteins between Reduced FT3 and Stable FT3 conditions.
Relies on the fold change > 1.5, numbers in bold in columns Red/CTRL and Stable/CTRL are referred to the dysregulated proteins in Reduced FT3 vs healthy controls and Stable FT3 vs healthy controls, respectively.
However, the overview of differential protein abundance among the three conditions (reduced FT3, Stable FT3, healthy controls) could be observed on the heatmap graph (Figure 5) that was modified according to the reviewer request to improve the understanding of differential proteome in the 3 considered conditions. We modified the figure 5 introducing coloured bars to better visualize the different clusters of protein abundance. The graph highlights a different trend of protein abundance between healthy controls (blue bar) and all thyroidectomised patients (orange bar). Moreover, thyroidectomised patients were in part subdivided into two other subgroups corresponding to reduced FT3 (yellow bar) and stable FT3 (purple bar).
We also add the following sentence to the discussion section. Page 14, line 292:
“In particular, CFAH, CFAB, coagulation factor XIII B chain (F13B), vitamin D binding protein (VTDB), APOA4, C1QB, APOAI, hemoglobin, C1S, A1AT fragments were up-regulated in FT3 reduced with respect to Stable FT3 patients while PLMN, CFAB frag-ments, histidine-rich glycoprotein and some APOAI fragments were down-regulated in FT3 reduced with respect to Stable FT3 patients.
On the other hand, serum down-regulated proteins in thyroidectomised patients then healthy controls were CFAH, alpha 1B glycoprotein, VTDB, APOAIV, C3 fragment C-term, C1S, APOAI fragments N-term, while up-regulated proteins were PLMN, CFAB full length and some fragments, serotransferrin, F13B, albumin fragments, APOAI fragments C-term, apolipoprotein E, and immunoglobulin heavy constant gamma 2.”
- This author concluded that LT4 replacement therapy might not restore euthyroid conditions in all thyroidectomized patients. This conclusion rise the question about how the influence of LT4 replacement therapy on participants with hypothyroidism who were not taking thyroidectomized therapy.
A18. The referee is right. Recent studies reported as proteomic profile in hypothyrois patients is different before and after replacement therapy with LT4 (Reference #6, #7, #8, #9).
In our study, the choice of thyroidectomized patients was mandatory in our study to verify the role of reduced FT3 values. These biochemical evidences are not frequently observed in hypothyroid patients in whom the presence of the thyroid is often sufficient to compensate for a possible reduced conversion of T to T4 at the peripheral level. Indeed, the majority of published studies, reduced FT3 levels were reported only in athyreotic patients.
We trust the paper is now in order for acceptance.
Yours sincerely,
Claudia Landi, Maria Grazia Castagna, Silvia Cantara and colleagues
Siena University, Italy
Round 2
Reviewer 3 Report
Thanks to this author’s effort, I thought present manuscript became improved. However, I still thought there are critical issue in present study. I commented as following.
- I’m still concerned that this author made case-control study with small number of control group. Normally, number of control group is same or higher than case group in case control study. However, in this study number of control group is smaller than that of case group. Therefore, I thought the reason why this author thought the number of 6 is enough for control in present study population should be clarified as I commented before. However, this author could not explain this reason in revised version of this manuscript. Since sample size of this study is small, there is tremendously high risk of influence of selection bias on present conclusion. Then verifying the reason why this author made the number of control group is smaller than that of case group. Case control study is designed to investigate the rare case of disease as you know. Then the effort to reduce the risk of selection bias in control group is mandatory. However, present study design could emphasize the influence of selection bias. Since this author did not follow ordinally method, this author should explain why number of 6 is (number of control group is smaller than case group) appropriate for control group in present study.
- In addition to that, another problem raised my concern. According to this author’s previous study, this author has a 140 case of thyroidectomized subjects. However, in present study the total number of thyroidectomized case is 16. Therefore, the risk of selection bias also could be influenced on case group.
- This author replied as following ”as reported in our previous paper (Ref. # 5), we define “reduced FT3” levels when the post-surgical FT3 levels was at least lower than 0.5 pg/ml respect to pre-surgical FT3 values. In our laboratory, the FT3 interassay CV is 8%. The difference of 0.5 pg/ml that we decided to be significant, was superior that 8% CV for each patient, suggesting that the reduction was clinically relevant.”. However, this sentence dose not holds enough evidence that the cut off point for 0.5 pg/mL is appropriate for present study. If there is study that intended to clarify the cut off point of this value, that could be appropriate as a reference. If not, efficiency of treatment should be compared within targeted study population and that value could not expanded for other study. If present study uses exactly same population with their previous study, cut off point for 0.5 pg/mL could holds enough evidence. However, in that case, the number of thyroidectomized cases could not holds enough value.
Author Response
Thanks to this author’s effort, I thought present manuscript became improved. However, I still thought there are critical issue in present study. I commentedasfollowing.
- I’m still concerned that this author made case-control study with small number of control group. Normally, number of control group is same or higher than case group in case control study. However, in this study number of control group is smaller than that of case group. Therefore, I thought the reason why this author thought the number of 6 is enough for control in present study population should be clarified as I commented before. However, this author could not explain this reason in revised version of this manuscript. Since sample size of this study is small, there is tremendously high risk of influence of selection bias on present conclusion. Then verifying the reason why this author made the number of control group is smaller than that of case group. Case control study is designed to investigate the rare case of disease as you know. Then the effort to reduce the risk of selection bias in control group is mandatory. However, present study design could emphasize the influence of selection bias. Since this author did not follow ordinally method, this author should explain why number of 6 is (number of control group is smaller than case group) appropriate for control group in present study.
We introduce in our analysis 6 Control subjects that we have in our databank that matched for age and gender and with no history of concomitant pathologies and not on any therapy in order to have a groupas homogeneous as possible with the other conditions.Our experimental design focused on the main differences across the studied cohorts thus we opted for a further threshold based on +/-1.5 FC in addition to the canonical statistical significance. Our choice would result in a theoretical increase of lost information in favour of more robust and highly informative biological meaning (less list, more sense). For this reason, we concentrated our efforts avoiding the type I error through BH-FDR. However, about type II error, the power analysis based on the our 57 spots emphasyzed that, using 8 and 6 samples per group, the power was higher than 0.8 for almost all the variable (just 3 had a power between higher than 0.7 and one higher than 0.6). Thus an increase of samples would not drastically improve our results.
As we mentioned at page 6, our overall quality control step was based on unsupervised multivariate analysis on the most variant spots. We observed that the general variance explanation was mainly related with the FT-levels and, in parallel, we could exclude any technical bias or covariant disturbances.
However, we report the necessity to validate our results on a large cohort of samples in the Limitations paragraph (Page 15, line 377):
“Limitations: Further studies including a large cohort of patients, will be needed to confirm our results. Moreover, a serum proteomic comparison between pre- and post-thyroidectomy in the same patients could be necessary. In order to validate our results, the level of inflammatory molecules to peripheral level, the coagulation and fibrinolytic state of the patients as well as the lipoprotein particle functionality, should be analysed”.
- In addition to that, another problem raised my concern. According to this author’s previous study, this author has a 140 case of thyroidectomized subjects. However, in present study the total number of thyroidectomized case is 16. Therefore, the risk of selection bias also could be influenced on case group.
In proteomic analysis, especially when applied two dimensional electrophoresis the number of the samples can’t be too higher to perform the experiments and for the image analysis, in order to do not make experimental errors.Number of samples have been chosen to perform a unique experiment in order to have the reproducibilityas higher as possible. For this reason, differences in protein spots that we observed are considered on spots that are present in all samples excluding possible variations and satisfying statistical analysis that take into consideration avoiding type 1 errors through the postHoc correction.
Normally, gel-based proteomic analysis are not performed on higher number of samples
EX:
Alfadda et al in Identification of Protein Changes in the Urine of Hypothyroid Patients Treated with Thyroxine Using Proteomics Approach on ACS Omega report a proteomic study on 9 subjects PMID: 33521475
Alfadda et al “Differences in the Plasma Proteome of Patients with Hypothyroidism before and after Thyroid Hormone Replacement: A Proteomic Analysis” published onInt J MolSciPMID: 29301248 report a proteomic study on 10 subjects.
- This author replied as following ”as reported in our previous paper (Ref. # 5), we define “reduced FT3” levels when the post-surgical FT3 levels was at least lower than 0.5 pg/ml respect to pre-surgical FT3 values. In our laboratory, the FT3 interassay CV is 8%. The difference of 0.5 pg/ml that we decided to be significant, was superior that 8% CV for each patient, suggesting that the reduction was clinically relevant.” However, this sentence dose not holds enough evidence that the cut off point for 0.5 pg/mL is appropriate for present study. If there is study that intended to clarify the cut off point of this value, that could be appropriate as a reference. If not, efficiency of treatment should be compared within targeted study population and that value could not expanded for other study. If present study uses exactly same population with their previous study, cut off point for 0.5 pg/mL could holds enough evidence. However, in that case, the number of thyroidectomized cases could not holds enough value.
The present study uses exactly same population with our previous study. In detail, we included in the proteomic analysis a small subgroup of patients that was representative of the whole cohort included in the previous study.
Nevertheless, we agree with the referee concerns and we report the necessity to validate our results on a large cohort of samples in the Limitations paragraph (Page 15, line 377):
“Limitations: Further studies including a large cohort of patients, will be needed to confirm our results. Moreover, a serum proteomic comparison between pre- and post-thyroidectomy in the same patients could be necessary. In order to validate our results, the level of inflammatory molecules to peripheral level, the coagulation and fibrinolytic state of the patients as well as the lipoprotein particle functionality, should be analysed”.